# Tolerability and Shared Decision-Making in the Hormonal Management of Endometriosis-Associated Pain

**DOI:** 10.3390/biomedicines13092294

**Published:** 2025-09-18

**Authors:** Diogo Pinto da Costa Viana, Leonardo Jacobsen, Igor Padovesi, Ana Comin, Eline Lobo de Souza Correia, Daniela Da Maia Fernandes, Ana Carolina Pires Dias

**Affiliations:** 1Brazilian Society of Endocrinology and Metabolism in Sports and Exercise, Florianópolis 88070-800, Brazil; 2Sociedade Brasileira de Pesquisa e Ensino em Medicina, São Paulo 01318-901, Brazilendo.dani@gmail.com (A.C.);; 3Sociedade Brasileira de Medicina da Obesidade, Florianópolis 88075-010, Brazil

**Keywords:** endometriosis, hormone replacement therapy, gonadotropin-releasing hormone, progestins, gestrinone, leuprolide, hormone antagonists, pelvic pain

## Abstract

**Background**: The management of endometriosis-associated pain has traditionally focused on analgesic efficacy. However, with high-level evidence demonstrating therapeutic equivalence among principal hormonal classes, the paradigm has shifted towards a patient-centred approach prioritising long-term tolerability and shared decision-making. Objectives: This review critically synthesises the evidence for the three main hormonal therapies—gonadotropin-releasing hormone (GnRH) analogues, dienogest, and gestrinone—focusing on their distinct tolerability and safety profiles to inform this modern clinical framework. **Methods**: This narrative review followed the SANRA (Scale for the Assessment of Narrative Review Articles) guidelines. The literature search was performed in PubMed, Embase, and Web of Science in June 2025. **Results**: Our comparative analysis, based on a structured literature search adhering to SANRA guidelines, shows that while all three classes are effective, they present distinct benefit–risk profiles: GnRH analogues offer potent pain relief but induce a hypoestrogenic state requiring add-back therapy to mitigate bone loss and vasomotor symptoms; dienogest preserves bone mineral density but is associated with challenging bleeding patterns and potential mood disturbances; gestrinone provides robust efficacy with a favourable cardiovascular and skeletal safety profile, although its androgenic effects can significantly impact patient adherence. **Conclusions**: In the absence of a clear hierarchy of efficacy, the optimal therapeutic choice is not determined by potency, but by a collaborative process in which patient values and tolerance for specific adverse effects guide selection. This review provides a framework to facilitate this shared decision-making (SDM) in clinical practice.

## 1. Introduction

Endometriosis is a chronic, oestrogen-dependent inflammatory disorder characterised by the presence of endometrial-like tissue outside the uterine cavity. It affects approximately 10% of women of reproductive age worldwide, with prevalence estimates ranging from 7% to 15% depending on population and diagnostic criteria [1,2,3]. This condition impacts more than 190 million individuals globally and is associated with substantial impairment in health-related quality of life (HRQoL) across physical, psychological, sexual, and occupational domains [1,4,5]. A multicentre study conducted across 10 countries involving 1418 women reported an average diagnostic delay of 6.7 years, with affected individuals losing an average of 10.8 h of work per week due to symptoms and experiencing significantly reduced physical functioning compared to controls [4]. Another cross-sectional analysis of women with laparoscopically confirmed endometriosis found an average weekly loss of 7.4 h of productivity and reported that presenteeism and pain-related impairment reached levels as high as 65% [5]. A recent umbrella review of systematic reviews further confirmed the strong association between endometriosis and multiple dimensions of psychological distress, including depression, anxiety, and deteriorated interpersonal relationships [6]. Furthermore, endometriosis frequently coexists with adenomyosis, a condition with overlapping symptoms and hormonal sensitivities, which further complicates the clinical management of associated pain and represents a target for the same hormonal suppression strategies.

The pathophysiology of endometriosis-associated pain involves a complex hormonal-inflammatory axis characterised by local oestradiol excess due to aberrant aromatase expression and deficient 17β-hydroxysteroid dehydrogenase type 2 (17β-HSD2) activity, combined with progesterone resistance mediated by selective downregulation of the PR-B isoform. These alterations promote chronic inflammation, neuroangiogenesis, and nociceptive sensitization [7,8,9,10,11]. These mechanisms provide the biological rationale for hormonal suppression therapy and are discussed in detail in Section 2.

Gonadotropin-releasing hormone (GnRH) analogues suppress ovarian oestradiol synthesis via hypothalamic–pituitary–axis inhibition, while progestins such as dienogest and gestrinone exert both central and local effects, including antagonism of oestrogenic activity and modulation of progesterone resistance [8]. Although mechanistically distinct, these therapeutic classes share a unified objective: disruption of the oestrogen-driven inflammatory environment that sustains the pathophysiology of endometriosis.

The aim of the present review is to critically compare the three main hormonal strategies in endometriosis, not only in terms of analgesic efficacy but, crucially, with respect to their safety profiles and tolerability. In light of mounting evidence suggesting therapeutic equivalence in efficacy, the decision-making process has shifted towards a more individualised approach, prioritising side-effect profiles and patient preferences as core determinants of clinical success. This review is not intended solely as a pharmacological overview, but rather as a patient-centred decision framework. By comparing the tolerability and safety profiles of GnRH analogues, dienogest, and gestrinone, we propose a structured model in which treatment choice is guided not by marginal differences in efficacy, but by the alignment between individual adverse-effect profiles and patient values.

Although other hormonal approaches, such as combined oral contraceptives or the levonorgestrel intrauterine system, are used in practice, these modalities were not the primary focus of this review, as high-quality comparative evidence on tolerability and long-term safety in endometriosis-associated pain is scarce. By contrast, GnRH analogues, dienogest, and gestrinone have been consistently evaluated in randomised trials and systematic reviews, making them the most appropriate classes for a structured comparative analysis. Unlike prior systematic and Cochrane reviews, which have primarily focused on efficacy outcomes, this review adds novelty by integrating tolerability profiles with a structured shared decision-making (SDM) framework, thereby offering clinicians a more practical, patient-centred approach to therapeutic decision-making.

## 2. Pathophysiology of Endometriosis-Associated Pain

Pain in endometriosis arises from a multifactorial interplay of hormonal, inflammatory, and neurological mechanisms within the lesion microenvironment. The disease is characterised by local oestrogen dominance and progesterone resistance, which together sustain chronic inflammation and pelvic hyperalgesia. Aberrant aromatase expression and deficient 17β-hydroxysteroid dehydrogenase type 2 (17β-HSD2) activity promote local oestradiol excess, driving proliferation of ectopic endometrial tissue and perpetuating prostaglandin-mediated inflammation [12,13,14].

Progesterone resistance, largely due to the loss of progesterone receptor isoform B (PR-B), further disrupts endometrial homeostasis and enhances inflammatory signalling. In parallel, inflammatory cytokines and oestrogens stimulate neuroangiogenesis, with increased expression of neurotrophic factors such as nerve growth factor (NGF), contributing to neural remodelling, nociceptor sensitisation, and the chronic pain phenotype of the disease [5,15,16].

These pathophysiological features provide the biological rationale for hormonal suppression therapy. By targeting oestrogen production or counteracting its local effects, agents such as GnRH analogues, dienogest, and gestrinone aim to disrupt this endocrine–inflammatory cycle and thereby alleviate pain.

## 3. Materials and Methods

This narrative review was conducted with the objective of critically synthesising the current evidence on the clinical efficacy, tolerability, and safety profiles of the three principal hormonal strategies used in the treatment of endometriosis-associated pain: dienogest, gonadotropin-releasing hormone (GnRH) analogues, and gestrinone. To ensure methodological transparency and clarity of clinical reasoning, this review followed the core principles outlined in the SANRA (Scale for the Assessment of Narrative Review Articles) guidelines.

A comprehensive literature search was performed in June 2025 across the MEDLINE/PubMed, Embase, and Web of Science databases. Distinct search strategies were developed for each therapeutic class, combining Medical Subject Headings (MeSH) with free-text keywords. See Table 1.

Eligibility criteria included studies of women of reproductive age with clinically or surgically confirmed endometriosis. Studies had to evaluate one of three hormonal treatments as monotherapy and report outcomes on pain relief, adverse effects, bone mineral density, or metabolic safety. Exclusion criteria were studies addressing infertility or assisted reproductive technologies, publications not in English, preclinical models, narrative reviews without systematic methodology, and studies without outcomes relevant to tolerability or long-term safety.

Study selection focused on the main themes of this review: efficacy, tolerability, and safety. Preference was given to systematic reviews, meta-analyses, and large randomised clinical trials when available.

For each included study, data were extracted on drug class, treatment duration, clinical outcomes, efficacy findings, adverse event profiles, and effects on bone and metabolic parameters. Findings were grouped by therapeutic class to allow structured comparison and to highlight tolerability differences that may influence adherence and patient satisfaction in real-world settings. Although no quantitative synthesis was performed, emphasis was placed on the clinical relevance of each finding and its implications for shared decision-making in gynaecologic endocrinology. A structured narrative format was chosen instead of a systematic review model to integrate diverse evidence into a cohesive and clinically meaningful framework for individualised treatment.

## 4. Comparative Analysis of Hormonal Therapies

### 4.1. Gonadotropin-Releasing Hormone (GnRH) Analogues

GnRH analogues constitute one of the most established and potent classes of pharmacological therapy for the management of moderate to severe endometriosis-associated pain. Their mechanism of action relies on suppression of the hypothalamic–pituitary–ovarian axis, thereby inducing a reversible state of chemical menopause. GnRH agonists, such as leuprolide and goserelin, act by initially stimulating and then downregulating pituitary GnRH receptors (resulting in the characteristic flare-up phenomenon), whereas oral GnRH antagonists, such as elagolix and relugolix, provide immediate, dose-dependent suppression of gonadotropin secretion without an initial stimulatory phase [9].

The efficacy of GnRH analogues is well established. In a pivotal randomised, double-blind, placebo-controlled trial, leuprolide acetate significantly reduced dysmenorrhoea and non-menstrual pelvic pain over a six-month treatment period [8]. The introduction of oral antagonists further strengthened this evidence base. Phase III randomised controlled trials, such as Elaris EM-I and EM-II, demonstrated that elagolix significantly reduced both menstrual and non-menstrual pelvic pain in a dose-dependent manner compared to placebo [17]. A recent Cochrane Reviews positioned leuprolide among the most effective pharmacological interventions for endometriosis-related pain [18,19,20].

Despite their robust efficacy, the profound hypoestrogenism induced by GnRH analogues is associated with a notable adverse effect profile, which constitutes the principal limitation of their long-term use and a major consideration in shared decision-making (SDM). Common side effects include vasomotor symptoms (e.g., hot flushes), vaginal dryness, mood changes, and loss of bone mineral density (BMD). RCT data have shown that monotherapy with leuprolide can lead to more than 6% loss of lumbar spine BMD within six months [21]. When compared directly with dienogest in a head-to-head randomised controlled trial, GnRH analogues exhibited comparable analgesic efficacy but were associated with a significantly higher incidence of hot flushes and a measurable decrease in BMD, whereas dienogest maintained BMD neutrality [22,23].

To mitigate these adverse effects and extend the duration of treatment, the strategy of add-back therapy has been developed. This approach is based on the oestrogen threshold hypothesis, which was originally proposed in an expert consensus and supported by RCTs [24]. Seminal randomised studies have demonstrated that add-back therapy prevents BMD loss and relieves hypoestrogenic symptoms without compromising the analgesic efficacy of GnRH analogues [16]. This concept has been incorporated into modern oral combination regimens. In the SPIRIT 1 and 2 phase III RCTs, relugolix combined with oestradiol and norethindrone acetate in a once-daily oral formulation significantly reduced pain while preserving bone density over a 12-month treatment period [25].

GnRH analogues, therefore, represent a highly effective pharmacological option for pain management in endometriosis. However, their use requires careful counselling regarding the adverse effects associated with hypoestrogenism. For extended treatment durations, add-back therapy is strongly recommended to ensure safety and tolerability. In SDM discussions, the hypoestrogenic burden and need for add-back therapy should be clearly addressed, allowing patients to weigh rapid analgesia against the risks of vasomotor symptoms and bone loss.

### 4.2. Dienogest

Dienogest is a synthetic oral progestin derived from 19-nortestosterone, developed specifically for the treatment of endometriosis and currently considered a first-line therapy in many international guidelines (expert consensus). Its mechanism of action is multifactorial: it exerts a strong progestogenic effect on the endometrium, resulting in decidualisation and subsequent atrophy of ectopic endometrial tissue. In addition, dienogest has local anti-inflammatory, antiangiogenic, and antiproliferative properties [26]. Unlike GnRH analogues, it induces only moderate ovarian suppression and does not cause profound hypoestrogenism [27].

The efficacy of dienogest at a daily dose of 2 mg has been well demonstrated in multiple randomised controlled trials (RCTs). In pivotal studies, it was shown to be significantly superior to placebo in reducing endometriosis-associated pelvic pain (EAPP), with clinically relevant improvements in pain scores observed at both 12 and 24 weeks of treatment [27,28].

A key argument supporting the clinical positioning of dienogest derives from direct head-to-head comparisons. A multicentre non-inferiority randomised controlled trial directly compared dienogest with leuprolide acetate (a GnRH analogue) over a 24-week period. The study concluded that dienogest was non-inferior to leuprolide in reducing pain, with nearly identical reductions in Visual Analogue Scale (VAS) scores at the end of treatment (−47.5 mm for dienogest versus −46.0 mm for leuprolide) [22]. This finding of equivalent analgesic efficacy was subsequently corroborated by network meta-analyses, which ranked dienogest at the same efficacy level as GnRH analogues for the treatment of endometriosis-related pain [19].

The principal advantage of dienogest, and a central theme in shared decision-making, lies in its distinct tolerability profile. Because it does not induce severe hypoestrogenism, dienogest is not associated with bone mineral density (BMD) loss. The same RCT that demonstrated efficacy equivalence also found that while the leuprolide group experienced a significant decrease in BMD, the dienogest group maintained stable bone density, positioning it as a safe option for long-term use without the need for add-back therapy [24]. Moreover, the incidence of vasomotor symptoms such as hot flushes is markedly lower when compared to GnRH analogues.

Nevertheless, dienogest presents its own set of tolerability challenges, particularly in relation to bleeding pattern disturbances, mood alterations, and weight gain. RCTs and pooled clinical trial analyses have shown that irregular bleeding, spotting, and amenorrhoea are among the most commonly reported adverse events and are frequent causes of treatment discontinuation [28,29]. Other progestogenic side effects, such as headache, breast tenderness, depressed mood, and acne, may occur, although generally with lower intensity [28].

A notable concern is the psychiatric safety profile of dienogest, especially the potential for depressive symptoms. Clinical studies indicate that up to 5% of women using dienogest may experience mood changes, typically mild to moderate in intensity [30]. However, more severe cases have also been reported, including major depressive disorder with suicidal ideation in patients without prior psychiatric history, as documented in case reports [31]. A large prospective observational pharmacovigilance study (VIPOS) reported a modest but measurable increase in the risk of depression among dienogest users, with an adjusted hazard ratio of up to 1.8 compared to other hormonal therapies for endometriosis [32]. Mechanistically, these mood-related effects may be mediated by changes in neurosteroid levels, particularly allopregnanolone, and its action on GABA-A receptors in limbic structures such as the hypothalamus and amygdala [33,34]. Importantly, while the VIPOS demonstrated a relative increase in depression risk, the absolute incidence remained low, with the majority of users not experiencing psychiatric events.

In light of these findings, it is advisable to conduct psychiatric screening prior to initiating treatment and to closely monitor mood symptoms, especially during the initial months of therapy.

Dienogest thus represents a therapeutic option with analgesic efficacy comparable to potent GnRH analogues, while offering the crucial benefit of preserving bone mineral density and avoiding severe vasomotor symptoms. Its use should involve a frank discussion with the patient regarding the acceptability of bleeding pattern disturbances as the principal adverse effect to be anticipated and managed. Within SDM, counselling on dienogest should emphasise its favourable bone safety compared to GnRH analogues, while also addressing the acceptability of irregular bleeding and the small but measurable risk of mood alterations, so that patients can make an informed choice based on their personal tolerance thresholds.

### 4.3. Gestrinone

Gestrinone is a synthetic steroid derived from 19-nortestosterone, pharmacologically classified as an antiprogestogen with antiestrogenic and moderate androgenic activity, due to its affinity for progesterone, androgen, and oestrogen receptors [35]. Its mechanism of action in endometriosis is multifaceted: it suppresses pituitary gonadotropin release, resulting in anovulation and a hypoestrogenic state [36], while simultaneously exerting a direct atrophic effect on ectopic endometrial tissue [37]. This pharmacological profile positions gestrinone as a high-potency hormonal treatment.

The analgesic efficacy of gestrinone is robust and well documented. A recent Cochrane overview concluded that, for the management of non-menstrual pelvic pain, gestrinone was the most effective intervention among all pharmacological treatments evaluated [19]. High-quality evidence supporting the central argument of this review comes from a multicentre, randomised, double-blind trial that directly compared gestrinone with leuprolide acetate (a GnRH analogue). The study demonstrated that gestrinone was equally effective in reducing pelvic pain, with no statistically significant difference between the treatment groups [38]. The clinical efficacy of gestrinone has also been recognised in Cochrane systematic reviews, which list it as a valid therapeutic alternative, albeit with a distinct side-effect profile [19].

A key consideration in therapeutic decision-making with gestrinone is its unique safety profile. Most importantly, its use has not been associated with serious adverse outcomes. A comprehensive systematic review of randomised and observational clinical trials, including 32 studies, found no reports of major cardiovascular events such as myocardial infarction or stroke, nor any mortality related to treatment [33]. Furthermore, in contrast to GnRH analogues, multiple RCTs and cohort studies have shown that gestrinone does not induce bone mineral density loss; in some cases, a slight increase or maintenance of BMD was observed during treatment [38,39].

The most frequently reported adverse effects of gestrinone are androgenic. RCTs, systematic reviews, and expert guidelines from the European Society of Human Reproduction and Embryology (ESHRE) consistently identify seborrhoea, acne, and hirsutism as the most common treatment-related events. These effects are dose-dependent, as shown in trials that compared varying gestrinone regimens. While often categorised as dermatological, these manifestations can significantly impact a patient’s quality of life and self-esteem, representing a critical potential barrier to treatment adherence. It is therefore important to note that individual susceptibility plays a significant role, making a personalised risk–benefit discussion essential in clinical decision-making.

Amenorrhoea, often induced by gestrinone, should not be regarded as an adverse effect in this context but rather as a therapeutic objective, since menstrual suppression is directly associated with relief from cyclic dysmenorrhoea [33].

Gestrinone emerges as a therapeutically valid option with analgesic efficacy comparable to, and in the case of non-menstrual pain, potentially superior to that of other leading therapies. Its favourable cardiovascular and skeletal safety profiles further enhance its clinical appeal. However, the primary limitation, which heavily influences its acceptability for many patients, lies in its androgenic effects. These manifestations, while dose-dependent and influenced by individual sensitivity, can be highly distressing and must be positioned as a central topic of an open dialogue with the patient during therapeutic planning. Within an SDM framework, clinicians should clearly present the trade-off between gestrinone’s strong analgesic efficacy and its androgenic side effects, allowing patients to decide whether the potential cosmetic and lifestyle impacts are acceptable in light of its favourable bone and cardiovascular safety profile.

### 4.4. Comparative Summary of Efficacy and Safety

High-quality scientific literature, including randomised controlled trials and network meta-analyses, consistently supports the conclusion that GnRH analogues, dienogest, and gestrinone demonstrate similar clinical benefit in alleviating endometriosis-associated pain [16,21,22]. Although subtle differences exist, such as the reported superiority of gestrinone for non-menstrual pelvic pain [21], the overarching clinical message is the absence of a clear hierarchy in analgesic potency. This paradigm is summarised in Table 2.

GnRH analogues offer potent pain relief but impose a hypoestrogenic burden, including vasomotor symptoms and, most importantly, significant bone loss, which mandates the addition of oestrogen–progestin supplementation for long-term use [17,24]. Dienogest achieves therapeutically equivalent outcomes while preserving bone mineral density (BMD), making it safer for extended therapy, although bleeding irregularities remain a key adherence challenge [16,22]. Gestrinone, meanwhile, offers comparable pain relief, with the added benefit of a favourable bone and cardiovascular safety profile. Its adverse effects are predominantly androgenic and aesthetic in nature, and their incidence depends largely on dosage and individual susceptibility [26,30]. As illustrated in Figure 1, these agents demonstrate equivalent efficacy, with differences in tolerability profiles guiding therapeutic choice [26,30].

Overall, there is no universally superior hormonal therapy for endometriosis-associated pain. Superiority is relative and patient-specific, and therapeutic choice must balance efficacy, safety, comorbidities, reproductive goals, and lifestyle factors. The following section will expand on how these considerations are integrated within a structured shared decision-making framework, which is increasingly recognised as essential to patient-centred care in endometriosis.

## 5. Discussion: Tolerability and Compliance

In the context of hormonal therapy for endometriosis, tolerability should be distinguished from adverse events. Adverse events are objectively recorded side effects in clinical trials, whereas tolerability reflects the patient’s subjective capacity to live with those effects without abandoning treatment, integrating perceived intensity, day-to-day impact, and acceptability in light of personal priorities and quality-of-life goals. This distinction is central to patient-centred care in endometriosis, where symptom burden and health-related quality of life are heavily affected and inherently shape treatment preferences and expectations [1,2,3,4,5,6]. In practice, a regimen may yield frequent adverse events yet remain well tolerated if effects are mild, manageable, or acceptable to the patient. Conversely, events that are uncommon but cosmetically or functionally disruptive can render a therapy poorly tolerated and jeopardise adherence [3,19,40].

This separation has direct implications for clinical decision-making across drug classes. With GnRH analogues, vasomotor symptoms, vaginal dryness, mood changes, and bone mineral density loss are expected consequences of hypoestrogenism and often drive the need for mitigation strategies or early discontinuation in routine care [3,17,21]. The oestrogen-threshold hypothesis provides the rationale for add-back regimens that maintain oestradiol within a therapeutic window to preserve bone and relieve vasomotor symptoms without reactivating lesions, thereby improving tolerability and extending safe use [24,25]. For dienogest, randomised and long-term clinical data show comparable pain relief with bone neutrality, which favours continuation, but tolerability is frequently threatened by irregular bleeding and mood symptoms; pharmacovigilance evidence from VIPOS suggests a measurable association with depression, underscoring the need for prospective screening and follow-up of psychiatric symptoms in susceptible patients [26,27,28,29,31,32,33]. For gestrinone, androgenic manifestations such as acne, seborrhoea, and hirsutism are the principal adherence barriers, although trials and reviews document therapeutic benefit and skeletal safety; counselling must explicitly address cosmetic concerns that disproportionately influence acceptability and long-term use [19,34,36,37,38,39].

Real-world continuation is typically lower than in randomised trials and is closely linked to the interplay between side-effect profiles and patient-reported outcomes. Extension and observational data with oral GnRH antagonists indicate that similar clinical benefit can be maintained beyond the pivotal trial windows, but long-term success depends on strategies that mitigate hypoestrogenic effects and protect bone health [21,25]. For dienogest, clinical programmes and pooled safety analyses describe sustained clinical effectiveness in long-term use, while registries highlight discontinuation related to bleeding disturbances and mood changes, emphasising the importance of early expectation-setting and active management of predictable adverse effects [26,27,28,29,31]. For gestrinone, historical randomised and cohort experiences support robust symptom relief and an absence of treatment-related bone loss, yet adherence is largely determined by the patient’s tolerance for androgenic effects and the feasibility of dose adjustments when needed [19,34,36,37,38,39].

Programming for failure requires anticipatory planning for when and how to suspend or switch therapies once tolerability thresholds are crossed. With GnRH analogues, the need for time-limited courses without add-back and the potential to extend duration with combination regimens should be discussed at initiation, including triggers for reassessment such as vasomotor burden or bone health parameters [17,21,24,25]. With dienogest, clinicians should set explicit plans for managing bleeding pattern changes and for monitoring mood, given real-world signals from pharmacovigilance, while leveraging the agent’s bone neutrality to support long-term continuation where acceptable [26,27,28,29,31,32,33]. With gestrinone, counselling should prioritise candid discussion of androgenic risks, individualised dose strategies, and early follow-up focused on patient-reported cosmetic impact, which often governs persistence more than analgesic benefit [19,35,36,37,38,39]. Across classes, the most effective regimen is the one a woman is willing and able to use consistently over time, aligning similar clinical benefit with a tolerability profile that fits her values, comorbidities, and life circumstances [1,2,3,4,5,6,19,40]. The clinical implications of these findings are further illustrated in Box 1, which outlines preferred hormonal strategies for different patient phenotypes.

Box 1Preferred hormonal strategies according to clinical phenotypes in women with endometriosis-associated pain.Clinical phenotypes and preferred hormonal strategies in endometriosis-associated pain1—Adolescents and young womenDienogest and gestrinone are preferred due to their bone-sparing profile and avoidance of profound hypoestrogenism, allowing safe long-term use without the need for add-back therapy. GnRH analogues should be avoided as first-line agents because of their association with accelerated bone loss.2—Patients with poor tolerability to prior therapiesShared decision-making is essential. Dienogest is suitable for women intolerant to vasomotor symptoms or bone effects, while gestrinone may be considered if mood changes or bleeding disturbances were problematic. GnRH analogues remain an option for short-term use with add-back therapy in those requiring rapid relief.3—Women prioritising fertility preservationAll agents suppress ovulation during treatment, but reversibility differs. Dienogest and gestrinone are associated with quicker recovery of ovulatory cycles than GnRH agonists, which may be advantageous for women planning conception shortly after therapy.4—Patients particularly sensitive to cosmetic or androgenic effectsGnRH analogues or dienogest are preferable over gestrinone for women in whom acne, seborrhoea, or hirsutism would strongly impact quality of life and adherence.

## 6. Shared Decision-Making in Hormonal Therapy for Endometriosis

Shared decision-making (SDM) is increasingly recognised as a cornerstone of best practice in the management of chronic gynaecological conditions such as endometriosis, where treatment must be sustained over long horizons and tailored to multifaceted quality-of-life outcomes [1,4]. SDM is defined as a collaborative process in which clinicians and patients make health decisions together by combining the best available clinical evidence with the patient’s individual values, goals, and preferences. Its core principles include respect for patient autonomy, transparent communication of risks and benefits, elicitation of preferences and concerns, and agreement on a treatment plan that reflects both medical evidence and personal priorities [41,42].

In practical terms, SDM in hormonal therapy for endometriosis begins with clarifying the outcomes that matter most to the patient, including pain control, preservation of bone health, protection of mood, menstrual regulation, reproductive goals, and the acceptability of aesthetic or lifestyle impacts. Evidence on efficacy and side-effect profiles should be presented in absolute terms, with emphasis on trade-offs—for example, the hypoestrogenic burden of GnRH analogues versus the bleeding disturbances of dienogest, or the androgenic effects of gestrinone despite its favourable skeletal and cardiovascular profile. Preference elicitation can be facilitated through structured counselling sessions, supported by decision aids that provide balanced information in plain language [43], and clinician-facing instruments such as the OPTION scale, which objectively measures the extent to which patients are involved in decisions [42].

Challenges to effective SDM must also be acknowledged. Women with endometriosis often face significant pain, fatigue, and psychological distress, which may impair concentration and participation in complex decision-making. Long diagnostic delays can erode trust in medical professionals, while cultural stigma surrounding menstrual and sexual symptoms may reduce the likelihood of openly expressing preferences. Health literacy and unequal access to follow-up care may further constrain the process. These obstacles can be mitigated by distributing counselling across more than one encounter, using plain language and visual risk displays, encouraging the presence of a support person, and scheduling structured reassessments that allow adaptation of treatment as tolerability and patient priorities evolve. These steps can be operationalised in practice through a structured process, as illustrated in Figure 2, which provides a practical flowchart for clinicians. The integration of these elements is also visually summarised in Figure 1, which illustrates how the absence of a clear hierarchy of efficacy shifts the focus of clinical care toward tolerability profiles and patient preferences, reinforcing the central role of SDM in the management of endometriosis.

The integration of these elements is visually summarised in Figure 1, which illustrates how the absence of a clear hierarchy of efficacy shifts the focus of clinical care toward tolerability profiles and patient preferences, reinforcing the central role of SDM in the management of endometriosis.

## 7. Limitations

This review was conducted according to the principles of a structured narrative synthesis and adheres to the SANRA guidelines to ensure methodological clarity and transparency. Nevertheless, several limitations should be acknowledged. As a narrative review, this work does not include quantitative synthesis or formal meta-analytic comparisons. The conclusions drawn regarding therapeutic equivalence are based on a critical qualitative appraisal of high-level evidence, including randomised controlled trials and network meta-analyses [16,21,31,40]. While this approach allows for a clinically grounded interpretation of diverse data sources, it precludes the statistical pooling of effect sizes or formal assessments of heterogeneity.

Further limitations arise from the underlying evidence base. The heterogeneity of the primary studies in terms of design, populations, and treatment regimens limits direct comparability and generalizability across trials. Notably, no randomised clinical trial to date has directly compared all three drug classes within a single framework, meaning comparative conclusions rely on indirect analyses, which are inherently less robust than head-to-head comparisons. Finally, as with all literature-based reviews, the possibility of publication bias cannot be entirely ruled out, as studies with negative or inconclusive results may be underrepresented.

With regard to gestrinone, while much of the published evidence derives from clinical trials conducted in the 1980s and 1990s, these studies were methodologically robust and remain clinically relevant. They consistently demonstrated strong analgesic efficacy, particularly in non-menstrual pelvic pain, as well as a favourable safety profile with respect to bone mineral density and cardiovascular risk. Extrapolation of older gestrinone studies to contemporary clinical practice must be made cautiously, particularly in regions where the drug is no longer available or widely prescribed. Importantly, gestrinone continues to be widely used in real-world clinical practice, particularly in Latin America and Southern Europe, where its therapeutic role is well established and supported by decades of accumulated clinical experience. Rather than constituting a limitation, the historical nature of this evidence highlights a robust legacy of use that predates the modern emphasis on patient-reported outcomes and shared decision-making metrics. Future studies incorporating contemporary tools for quality-of-life assessment and adherence monitoring would serve to expand, not validate, the existing evidence base. Future research should not only expand the evidence base with modern metrics but also explore the regulatory and pharmacoeconomic factors that currently limit gestrinone’s availability in many regions, despite its long-standing clinical use and favourable safety profile.

A limitation of this review is that it did not explore surgical interventions, which remain an important therapeutic avenue in the management of endometriosis-associated pain. Evidence indicates that laparoscopic techniques and other gynecologic procedures can substantially affect pain outcomes and quality of life [44,45]. Future studies should better integrate medical and surgical perspectives to provide comprehensive guidance for long-term care.

Despite these limitations, the available evidence is sufficiently consistent and clinically compelling to support the central message of this review: GnRH analogues, dienogest, and gestrinone all offer comparable efficacy in the management of endometriosis-associated pain. Therapeutic choice should therefore be driven not by assumptions of superiority, but by patient-specific factors such as tolerability, safety, comorbidities, and personal preferences within a shared decision-making model.

## 8. Conclusions

The management of endometriosis-associated pain has shifted from the pursuit of a pharmacological agent with superior efficacy to a more patient-centred framework that prioritises long-term tolerability and quality of life. High-level evidence from clinical trials and systematic reviews demonstrates that GnRH analogues, dienogest, and gestrinone provide similar clinical benefits in terms of pain reduction for women with endometriosis [16,21,31,40]. In this context, therapeutic selection is no longer determined by relative potency, but by the adverse effect profile most compatible with the patient’s values, comorbidities, and life circumstances.

Each treatment class presents a distinct balance between benefits and risks. GnRH analogues are potent and effective but induce a chemical menopause with predictable hypoestrogenic side effects, including significant bone mineral density loss, necessitating the use of add-back therapy for extended treatment [17,27,35]. Dienogest achieves comparable pain relief while preserving bone integrity, but irregular bleeding and potential mood disturbances may affect adherence [30,31]. Gestrinone, with its robust analgesic profile and absence of documented cardiovascular or skeletal harm, offers an appealing alternative, although its androgenic side effects must be carefully discussed and monitored [19,24,40,46].

In the absence of a clear hierarchy of efficacy, the therapeutic decision should be guided by a model of shared decision-making. The role of the clinician is to present the available evidence transparently, delineating the differential side-effect profiles of each option, and to support the patient in selecting a treatment aligned with her personal priorities and tolerability thresholds. The most effective therapy is not merely the one that reduces pain in clinical trials, but the one that a woman can safely and comfortably adhere to over time in the context of her lived experience.

## Figures and Tables

**Figure 1 biomedicines-13-02294-f001:**
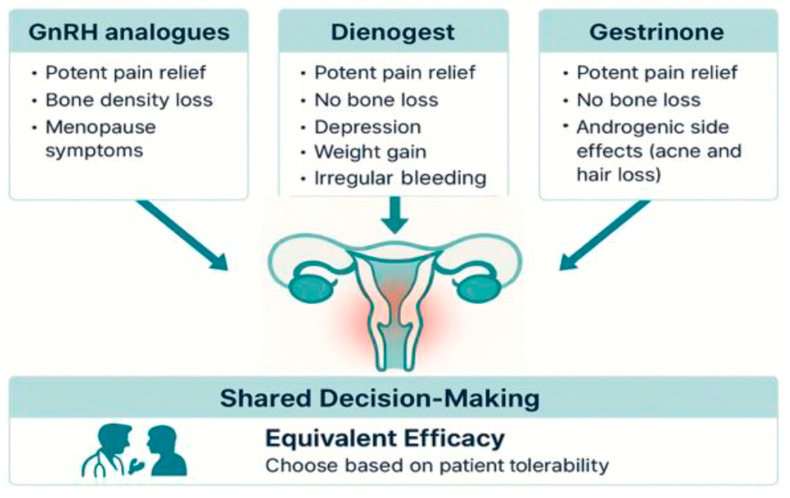
Comparative overview of three hormonal therapies for endometriosis-associated pain. All options offer equivalent efficacy in pain relief but differ in side effect profiles. Treatment choice should be guided by patient tolerability and shared decision-making.

**Figure 2 biomedicines-13-02294-f002:**
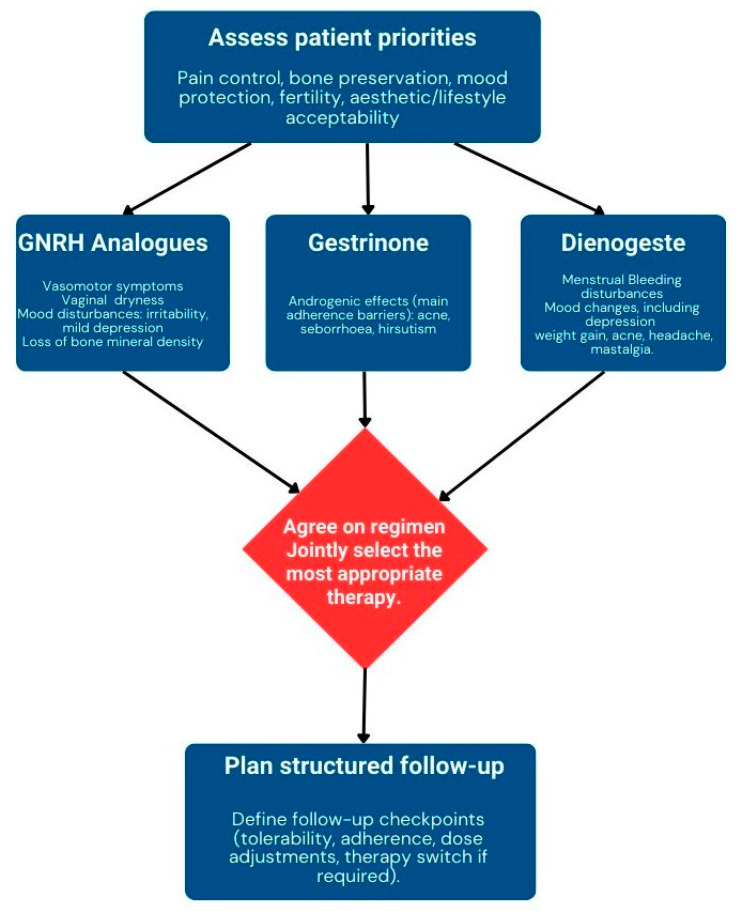
Shared decision-making framework for hormonal therapy in endometriosis. Flowchart illustrating the structured process of SDM: assessing patient priorities, presenting efficacy–tolerability trade-offs, eliciting preferences, agreeing on the most appropriate regimen, and planning structured follow-up.

**Table 1 biomedicines-13-02294-t001:** Search strategies used in PubMed, Embase, and Web of Science for studies on dienogest, GnRH analogues, and gestrinone in the management of endometriosis.

Dienogest/Search Strategy
Pubmed—(“Endometriosis” [MeSH] OR “Pelvic pain”) AND (“Progestins” [MeSH] OR “Dienogest”) AND (“Hormone Replacement Therapy” [MeSH] OR “Hormonal therapy” OR “Medical treatment”)
Embase—(‘endometriosis’/exp OR ‘pelvic pain’) AND (‘dienogest’/exp OR ‘progestinss’/exp) AND (‘hormone replacement therapy’/exp OR ‘hormonal therapy’ OR ‘medical treatment’)
Web of Science—TS = (“Endometriosis” OR “Pelvic pain”) AND TS = (“Dienogest” OR “Progestin”) AND TS = (“Hormonal therapy” OR “Medical treatment”)
GnRH analogues/Search Strategy
Pubmed—(“Endometriosis” [MeSH] OR “Endometriosis-related pain”) AND (“GnRH agonists” OR “Leuprolide” OR “Goserelin”) AND (“Hormonal therapy” OR “Medical management”)
Embase—(‘endometriosis’/exp OR ‘endometriosis-associated pain’) AND (‘gonadotropin-releasing hormone’/exp OR ‘gnrh agonists’) AND (‘leuprolide’ OR ‘goserelin’ OR ‘buserelin’)
Web of Science—TS = (“Endometriosis” OR “Pelvic pain”) AND TS = (“GnRH agonists” OR “Leuprolide” OR “Goserelin”) AND TS = (“Hormonal therapy” OR “Medical treatment”)
Gestrinone/Search Strategy
Pubmed—(“Endometriosis” [MeSH] OR “Pelvic pain”) AND (“Gestrinone” [MeSH] OR “Androgenic steroid”) AND (“Hormonal therapy” OR “Medical treatment”)
Embase—(‘endometriosis’/exp OR ‘pelvic pain’) AND (‘gestrinone’/exp OR ‘androgenic steroid’) AND (‘hormonal therapy’ OR ‘medical treatment’)
Web of Science—TS = (“Endometriosis” OR “Pelvic pain”) AND TS = (“Gestrinone” OR “Androgenic steroid”) AND TS = (“Hormonal therapy” OR “Medical treatment”)

**Table 2 biomedicines-13-02294-t002:** Comparative profile of hormonal therapies for endometriosis-associated pain.

Characteristic	GnRH Analogues (Agonists and Antagonists)	Dienogest	Gestrinone
Main Mechanism of Action	Deep suppression of the hypothalamic–pituitary–ovarian axis, inducing chemical menopause.	Progestin with strong endometrial effect and moderate ovarian suppression.	Antiprogestin with antiestrogenic and androgenic effects; pituitary suppression.
Efficacy on Pain	High efficacy, comparable to other options.	High efficacy, comparable to GnRH analogues.	High efficacy, comparable to other options; potentially superior for non-menstrual pelvic pain.
Main Advantage	Potency and rapid symptom suppression.	No bone loss, allowing long-term use without add-back therapy.	No bone loss and no severe hypoestrogenic effects.
Main Adverse Effect Profile	Hypoestrogenism.	Altered bleeding patterns.	Androgenic effects.
Effect on Bone Mineral Density (BMD)	Significant reduction; risk of osteopenia/osteoporosis with prolonged use.	Neutral effect; no BMD loss.	Neutral or slightly positive effect; no BMD loss.
Reported Cardiovascular Risk	Risk associated with premature menopause (long-term).	No evidence of increased risk.	No reports of negative cardiovascular outcomes in clinical trials.
Need for Add-Back Therapy	Yes, for use beyond 6 months to protect BMD and mitigate vasomotor symptoms.	No.	No.
Main Patient Complaints	Hot flashes, vaginal dryness, mood changes.	Irregular bleeding, spotting.	Acne, oily skin and hair, hirsutism, voice changes (rare).
Common Dosing Regimen	Monthly/quarterly injections (agonists) or daily pills (antagonists).	Daily oral tablet (2 mg).	Oral or vaginal capsules, 2–3 times per week.

## Data Availability

All data presented in this review are available in the cited references.

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
