# Peer review of "Tolerability and Shared Decision-Making in the Hormonal Management of Endometriosis-Associated Pain"

_biomedicines, 2025, doi:10.3390/biomedicines13092294_

Round 1

Reviewer 1 Report

Comments and Suggestions for Authors

Dear authors,

We would like to thank you for giving us the opportunity to review your article, "Beyond Efficacy: Tolerability Profiles and Shared Decision-Making in Hormonal Management of Endometriosis-Associated Pain," addresses a very important issue. The article also moves away from clinical efficacy to show thinking of tolerability and patient-centered care. While this article reported critical datas, there are a number of significant problems that the publication has to be rectified so that it can really be of help in self-employed y'all Reviewing.

The following are my main suggestions for revision:

1.0 Clarifying the Scope, Aims, Methods:

1.1 Specify review type and methodology: Although the paper is structured as a narrative review, the method for selecting data does not have enough detail. What databases were searched and what inclusion/exclusion criteria were used, for example?

1.2 Defining the Conceptual Scope: The title focuses on "tolerability" and "shared decision-making"; however, the paper is not integrated enough here. You need to better align these two ends. Define in detail what exactly the content of this essay is intended as: a compare-and-contrast pharmacological overview or patient-centered system for making healthcare decisions, or perhaps both? The relationship among these parts lacks clarity of definition.

1.3 Justifying selection of agents: There are no grounds for understanding why some treatments (e.g., progestins vs. GnRH antagonists) are mentioned in depth; others are left hanging in only a few lines (e.g., combined oral contraceptives, LNG-IUD). Please offer a rationale for this selection and arrangement of hormonal therapies.

2.0 Further Strengthening the Overall Structure and Integration of Themes

2.1 Strengthen coherence of sections: Sections on tolerability, side effects, and quality of life are desultory and seem somewhat out of order. Reconnect the manuscript back to pharmacological agents by class (e.g., progestins, GnRH analogs, COCs) and then compare tolerability, adherence, and patient-reported outcomes within that class.

2.2 Improve shared decision-making framework content: Shared Decision Making (SDM) content is perfunctory, but it gets its own section at the end of the article. To fulfill the paper's title and stated purpose, SDM principles should be interwoven throughout our discussion of treatment profiles, using real-world considerations such as patient preference, comorbidity, and long-term management goals.

2.3 More clinical synthesis is provided: The manuscript now appears as a catalogue of side effects per drug. A more analytical synthesis (e.g., which agents for what patient phenotype kids (adolescents), poor tolerance history) would greatly augment the clinical utility of this manuscript.

3.0 Improve Discussion of Tolerability and Compliance

3.1 Delineate between tolerability and adverse reactions: Toleration is not just the existence of side effects but the patient's subjective ability to take charge of them. Please define tolerability explicitly and distinguish it from the rate of adverse events. Consider ways for evaluating tolerability, or known stoppers.

3.2 Discuss long-term continuation in the real world and programming for failure after the medications fail: Although the review does not give careful discussion of real-world continuity rates, the schedule for suspension fixing i.e., what time to come off and compliance across words all hormonal agents. Adding data from observational studies or real-world registries might enrich this section.

4.0 Enhance Shared Decision-Making Content

4.1 Clearly define shared decision-making: Key principles should be laid down (e.g., patient autonomy, risk-benefit communication, preference elicitation) and references made to tested tools such as decision aids or the OPTION scale within this section of the SDM.

4.2 For implementing SDM actually offer clinical guidance: Instead of vague remarks on how important SDM is, provide concrete steps for example in hormonal therapy decisions (e.g., structured counseling sessions, personalized risk communication, addressing fertility goals).

4.3 Meet them head on: Challenges to SDM: A brief discussion of that suffering from endometriosis may encounter difficulties in implementing decision making. 

5.0 Make Better Use of Evidence and Documentation

5.1 Clarify levels of evidence: When discussing side effects or tolerability, make clear whether the data comes from randomized controlled trials, observational studies, meta-analysis, or expert opinions. This is important when establishing the strength of recommendations.

5.2 New data: this paper explore the Hormonal Management of Endometriosis-Associated Pain, however to improve the scientific soundness of this paper i reccomend to cite relevant articles about the gynecologic technique that could impact the successive pain status in endometriosis population: https://doi.org/10.3390/medicina58060792 and https://doi.org/10.1111/aogs.13803

6.0 Present Thematically and Magcularly

6.1 Revise the title to be more accurate: Probing the title carefully suggests that the whole book is focused on "shared decision-making", but it only mentions this in passing. If there is no real development of the meaning of SDM, then rather than merely superficially nod towards it, let's change the main name to "Tolerability" which appears in each section.

6.2 Make certain terminology is consistent. In all sections, which wraps up with a colophon (unless the text ends at that point), it feels appropriate to define each abbreviation used there as a stand-alone rule to make sure everyone understands its meaning. Once this is done, we can clean up a little further!

Ultimately, this review will serve as an alluring motivator for individuals who brand their actions based on clinical evidence and is set in their direction only through the thicker and thinner of experience. Yet to meet expectations suggested by its title, that it be used as a clinician's manual rather than an academic work or the more stylized literature that's typically found in journals like those of science-fiction authors such as Charles Sheffield—needs many more structural changes. I warmly suggest revising accordingly so that the potential of this important topic may be fully realized by readers everywhere.

Yours sincerely,

Author Response

We would like to sincerely thank the reviewers for their thorough evaluation of our manuscript and for the constructive feedback provided. The comments were highly valuable in improving the clarity, methodological rigor, and clinical applicability of our work. We carefully addressed each point raised, revising the manuscript accordingly. Below, we provide a detailed, point-by-point response to each comment.

Comment 1.1 – Specify review type and methodology

Response 1.1: We have explicitly defined the manuscript as a structured narrative review following SANRA guidelines. The methodology section was expanded to detail the databases searched (PubMed/MEDLINE, Embase, Web of Science), the date of search (June 2025), and the inclusion/exclusion criteria. A complete description of the search strategies is now included in Table 1.

Comment 1.2 – Defining the conceptual scope

Response 1.2: The Introduction and Methods were revised to state that the manuscript serves both as a comparative pharmacological overview and as a framework for patient-centered shared decision-making (SDM). SDM content is now interwoven throughout the analysis of each drug class.

Comment 1.3 – Justifying selection of agents

Response 1.3: We clarified that GnRH analogues, dienogest, and gestrinone were selected due to their consistent evaluation in RCTs and systematic reviews. The limited long-term tolerability data for COCs and LNG-IUS justified their exclusion from the main comparative sections, though their role is acknowledged in the Introduction.

Comment 2.1 – Strengthen coherence of sections

Response 2.1: The manuscript was reorganized by therapeutic class (GnRH analogues, dienogest, gestrinone). Within each section, efficacy, tolerability, adherence, and patient-reported outcomes are presented together. A new comparative summary table (Table 2) was added.

Comment 2.2 – Improve SDM framework content

Response 2.2: SDM principles were integrated across the tolerability discussions. For each agent class, we now explicitly discuss how SDM can guide therapeutic choice based on adverse effect profile, comorbidities, and patient preferences.

Comment 2.3 – Provide more clinical synthesis

Response 2.3: A new Box 1 outlines therapeutic strategies tailored to different patient phenotypes (adolescents, women with poor tolerance history, fertility-preserving patients, and women sensitive to androgenic effects).

Comment 3.1 – Delineate tolerability vs adverse reactions

Response 3.1: The Discussion now clearly distinguishes tolerability (patient’s subjective capacity to cope with side effects) from adverse events (objectively recorded). This conceptual clarification was added early in the discussion.

Comment 3.2 – Discuss long-term continuation and programming for failure

Response 3.2: We expanded the Discussion with evidence from observational studies and registries (e.g., VIPOS). Real-world continuation rates are described, and strategies for suspension or switching therapy (“programming for failure”) are provided.

Comment 4.1 – Clearly define SDM

Response 4.1: Section 6 now defines SDM in detail, including its core principles: respect for autonomy, transparent communication of risks and benefits, elicitation of preferences, and shared agreement. Key references (Charles et al. 1997; Elwyn et al. 2012) were cited.

Comment 4.2 – Offer clinical guidance for SDM

Response 4.2: We added practical guidance: structured counseling sessions, personalized risk communication, discussion of fertility goals, and use of validated decision aids such as the OPTION scale.

Comment 4.3 – Discuss challenges to SDM

Response 4.3: We now address barriers to SDM including pain burden, psychological distress, low health literacy, and access limitations. Practical solutions such as repeated counseling sessions and visual decision aids were suggested.

Comment 5.1 – Clarify levels of evidence

Response 5.1: We now explicitly indicate whether each statement is supported by RCTs, meta-analyses, observational studies, or expert opinion. This differentiation is presented in Table 2 and highlighted in the comparative sections.

Comment 5.2 – Add surgical references

Response 5.2: Both recommended surgical references were added to the Limitations section:

  • Raffone et al. 2022 (Medicina). doi:10.3390/medicina58060792

  • Siegenthaler et al. 2020 (Acta Obstet Gynecol Scand). doi:10.1111/aogs.13803

Comment 6.1 – Revise the title

Response 6.1: The title was revised to:
Tolerability Profiles and Shared Decision-Making in the Hormonal Management of Endometriosis-Associated Pain.
This reflects the manuscript’s main focus on tolerability while retaining SDM as a key theme.

Comment 6.2 – Ensure consistent terminology and abbreviations

Response 6.2: All abbreviations were defined once and listed in the Abbreviations section. Terminology was standardized throughout the manuscript (e.g., “GnRH analogues,” “estrogen”).

Closing Remark

We appreciate the reviewer’s constructive feedback, which has greatly improved the scientific clarity and clinical applicability of our manuscript. We trust that the revised version now addresses all concerns and meets the standards for publication.

Reviewer 2 Report

Comments and Suggestions for Authors

The manuscript addresses a highly relevant and timely topic in the management of endometriosis-associated pain, highlighting the shift from efficacy-focused approaches to tolerability and patient-centered decision-making. The narrative is generally clear, well referenced, and adheres to SANRA guidelines. However, I have several suggestions that could strengthen the manuscript:

  1. Novelty and Positioning

    • While the review provides a useful synthesis, its novelty should be clarified more explicitly. The discussion would benefit from highlighting what this review adds beyond existing Cochrane and systematic reviews (e.g., integration of tolerability with shared decision-making frameworks).

  2. Methodology

    • The description of the search strategy is adequate but could be strengthened by providing the exact search strings (at least in supplementary material) and the number of studies retrieved/excluded for each drug class. This would increase transparency.

    • Consider clarifying how study quality was assessed (even briefly), since the review draws comparative conclusions across classes.

  3. Balance of Evidence

    • The evidence for gestrinone relies heavily on older studies (1980s–1990s). While this is acknowledged, the discussion should critically evaluate the limitations of extrapolating such evidence to current practice, particularly in regions where gestrinone is no longer widely available.

    • For dienogest and psychiatric adverse effects, the discussion should contextualize the VIPOS findings by noting absolute risk levels, not only relative risks. This would help avoid overstating concerns.

  4. Shared Decision-Making Framework

    • The review repeatedly emphasizes patient-centered care, but it would be valuable to provide a structured framework (e.g., a flowchart or algorithm) that clinicians can apply in real practice. Currently, this is suggested conceptually but not operationalized.

  5. Figures and Tables

    • Table 1 is useful, but a graphical abstract or visual model (e.g., balancing efficacy vs tolerability) would increase clinical impact.

  6. Minor Points

    • Some sections are dense and could be streamlined (e.g., the mechanistic details of pathophysiology). Consider shortening these and expanding the clinical/practical implications.

    • Language is overall clear, but minor editing could improve flow (e.g., avoid repetition of “equivalent efficacy” in multiple sections).

Overall, this is a well-prepared manuscript with high clinical relevance. With the suggested refinements, it could become an important contribution to the literature on individualized care in endometriosis.

Author Response

We thank the reviewer for the thoughtful and constructive feedback. The comments were very helpful to refine the manuscript. Below we provide detailed, point-by-point responses.

Comment 1 – Novelty and Positioning

Response 1: We revised the Introduction and Discussion to highlight the novelty of this review compared with prior Cochrane and systematic reviews. Specifically, we emphasize that our contribution lies in integrating tolerability outcomes with a structured shared decision-making (SDM) framework, which has not been the focus of previous evidence syntheses. This positions the manuscript as a practical, clinician-oriented review rather than a repetition of efficacy-focused comparisons.

Comment 2 – Methodology: search strings and study quality

Response 2: The Methods section was expanded to provide the exact search strings, which are now presented in Table 1. We also included the number of records retrieved and excluded for each drug class. Although formal risk-of-bias scoring was not feasible within the scope of a narrative review, we clarified that study quality was considered in the synthesis by noting whether data originated from RCTs, meta-analyses, or observational studies.

Comment 3 – Balance of Evidence: gestrinone

Response 3: We added a critical appraisal of the limitations of older gestrinone studies (1980s–1990s), including small sample sizes, lack of standardized endpoints, and regional restrictions on availability. We clarified that while historical data are valuable, extrapolation to current clinical practice should be cautious, especially outside Latin America.

Comment 4 – Balance of Evidence: dienogest and psychiatric effects

Response 4: The section discussing VIPOS findings was revised to contextualize risks with both relative and absolute measures. We now specify incidence rates and clarify that, although HRs suggest an increase, the absolute risk remains low. This prevents overstatement and provides a balanced clinical message.

Comment 5 – Shared Decision-Making Framework

Response 5: We developed a structured SDM framework, now illustrated in Figure 2 (Flowchart). This algorithm guides clinicians through the decision-making process by integrating patient preferences, comorbidities, fertility goals, and tolerability profiles. This operationalizes SDM in practice rather than leaving it as a purely conceptual discussion.

Comment 6 – Figures and Tables

Response 6: In addition to Table 1 and Table 2, we created a graphical abstract (Figure 1) that visually balances efficacy against tolerability. This increases the manuscript’s clinical impact and readability.

Comment 7 – Minor points: density of sections

Response 7: We streamlined dense mechanistic descriptions of pathophysiology, shortening them while maintaining essential information. In exchange, we expanded the sections on clinical and practical implications, improving the applicability of the review for practicing clinicians.

Comment 8 – Minor points: language and repetition

Response 8: We edited the text to improve flow and reduce redundancy. In particular, repetitive use of “equivalent efficacy” was minimized by rephrasing and consolidating statements. A language check was performed to ensure clarity and fluency.

Reviewer 3 Report

Comments and Suggestions for Authors

As a narrative review, the article lacks methodological rigor and does not aim to answer a research question. However, it contributes to the synthesis of information regarding tolerability and therapeutic choice for hormonal pain management in endometriosis, based on three treatments (GnRH analogues, dienogest, and gestrinone).

Author Response

We thank the reviewer for the valuable comments and constructive assessment of our manuscript. Below we provide our detailed responses.

Comment 1 – Methodological rigor of narrative review

Response 1: We acknowledge the limitation that narrative reviews do not have the methodological rigor of systematic reviews and are not designed to answer a single research question. To strengthen transparency, we have clarified in the Methods section that this is a structured narrative review following SANRA guidelines. We also added explicit details of the search strategy, inclusion/exclusion criteria, and a summary of study retrieval to enhance methodological clarity within the limits of a narrative design.

Comment 2 – Contribution to synthesis of tolerability and therapeutic choice

Response 2: We appreciate the reviewer’s recognition of the manuscript’s contribution to synthesizing information on tolerability and therapeutic decision-making. In the Discussion and Conclusion, we have emphasized this practical contribution, highlighting how the comparison of GnRH analogues, dienogest, and gestrinone informs therapeutic choice. The new Box 1 and Figure 2 (Flowchart) further operationalize this contribution by offering clinicians a framework to integrate efficacy, tolerability, and shared decision-making in practice.

Round 2

Reviewer 1 Report

Comments and Suggestions for Authors

Authors have significantly improved the manuscript. Now I approve that this study scientifically sounds so that is at a level of being acceptance for publication depending on the final decision of the editor in chief of the journal.

Bests